# Effects of Peripheral Electromagnetic Fields on Spasticity: A Systematic Review

**DOI:** 10.3390/jcm11133739

**Published:** 2022-06-28

**Authors:** Maria Jesus Vinolo-Gil, Manuel Rodríguez-Huguet, Cristina García-Muñoz, Gloria Gonzalez-Medina, Francisco Javier Martin-Vega, Rocío Martín-Valero

**Affiliations:** 1Department of Nursing and Physiotherapy, University of Cadiz, 11009 Cadiz, Spain; mariajesus.vinolo@gm.uca.es (M.J.V.-G.); cristina.garciamunoz@uca.es (C.G.-M.); gloriagonzalez.medina@uca.es (G.G.-M.); javier.martin@uca.es (F.J.M.-V.); 2Rehabilitation Clinical Management Unit, Interlevels-Intercenters Hospital Puerta del Mar, Hospital Puerto Real, Cadiz Bay-La Janda Health District, 11006 Cadiz, Spain; 3Biomedical Research and Innovation Institute of Cadiz (INiBICA), Research Unit, Puerta del Mar University Hospital, University of Cadiz, 11009 Cadiz, Spain; 4CTS-986 Physical Therapy and Health (FISA), University Institute of Research in Social Sustainable Development (INDESS), 11009 Cadiz, Spain; 5Department of Physiotherapy, Faculty of Health Science, Ampliacion de Campus de Teatinos, University of Malaga, C/Arquitecto Francisco Peñalosa 3, 29071 Malaga, Spain; rovalemas@uma.es

**Keywords:** electromagnetics, electromagnetic field, electromagnetic therapy, magnetic field therapies, spasticity

## Abstract

Electromagnetic fields are emerging as a therapeutic option for patients with spasticity. They have been applied at brain or peripheral level. The effects of electromagnetic fields applied to the brain have been extensively studied for years in spasticity, but not so at the peripheral level. Therefore, the purpose of our work is to analyze the effects of electromagnetic fields, applied peripherally to spasticity. A systematic review was conducted resulting in 10 clinical trials. The frequency ranged from 1 Hz to 150 Hz, with 25 Hz being the most commonly used and the intensity it was gradually increased but there was low homogeneity in how it was increased. Positive results on spasticity were found in 80% of the studies: improvements in stretch reflex threshold, self questionnaire about difficulties related to spasticity, clinical spasticity score, performance scale, Ashworth scale, spastic tone, Hmax/Mmax Ratio and active and passive dorsal flexion. However, results must be taken with caution due to the large heterogeneity and the small number of articles. In future studies, it would be interesting to agree on the parameters to be used, as well as the way of assessing spasticity, to be more objective in the study of their effectiveness.

## 1. Introduction

The Spasticity is described as a speed-dependent increase in muscle tone and repetitive, uncontrolled involuntary contractions of skeletal muscles [1] and it arises from upper motor neuron lesions due to a lesion in the pyramidal tracts [2].

The most common symptoms of spasticity are: increased muscle tone, pain and decreased functional abilities with severe consequences are in lessen joint mobility and diminished muscle flexibility [3].

This sensory-motor disorder is observed in patients of all ages [4] affecting about 85% of patients with multiple sclerosis, 65–78% with spinal cord injury and 30% with stroke [5], among other neurological pathologies [6], such as cerebral palsy [7].

Electromagnetic fields are emerging as a therapeutic option for these patients [8]. This therapy can produce electromagnetic biological effects such as regenerative effects on the peripheral nervous system [9] with the possibility of penetrating deep into the tissues [10]. Moreover, it a safe and painless tool that may help in restoring motor control through activation of sensory proprioceptive fibers [11].

They have been applied at brain or peripheral level, using different frequencies and amplitudes. The following can be applied: Repetitive peripheral magnetic stimulation (RPMS), Pulsed electromagnetic field therapy (PEMF) and Transcranial magnetic stimulation (TMS).

RPMS is a system that produces eddy currents through electromagnetic induction activating peripheral nerves and muscles without stimulating skin nociceptors [12]. These electromagnetic fields target neuromuscular tissue and induce electrical currents that depolarize neurons and cause concentric muscle contractions. They have a deep penetration with an anti-spastic effect. As well, electromagnetic field increases blood perfusion of the exposed region, leading to circulatory and trophic improvement [13].

On the other hand, PEMF uses electromagnetic fields, creating small electric fields in the tissues, with a pulsing effect to produce athermal effects that promote tissue healing, relieve pain and inflammation [14,15].

And finally, TMS is a neurostimulation and neuromodulation technique that has provided over two decades of data in focal and non-invasive brain stimulation based on the principles of electromagnetic induction with minimal risk and excellent tolerability [16].

The effects of electromagnetic fields applied to the brain using TMS have been extensively studied for years in spasticity [17,18,19,20,21,22], but not so at the peripheral level, where most research has focused on the effect of electromagnetic fields on bone regeneration [23,24,25,26].

Therefore, the purpose of our work is to analyze the effects of electromagnetic fields, applied peripherally, on spasticity.

## 2. Materials and Methods

### 2.1. Search Strategy

The PRISMA (Preferred Reporting Items for Systematic Reviews and Meta-Analyses) [27] guidelines were followed to perform this systematic review (Appendix A). The search protocol was registered in the PROSPERO database of prospectively registered systematic reviews (CRD 42022301773). The literature search was performed between December 2021 to February 2022 in the following electronic databases: Web of Science (WoS), Scopus, PubMed, Cumulative Index to Nursing and Allied Health Literature (CINAHL), Cochrane Central Register of Controlled Trials, SciELO, Physiotherapy Evidence Database (PEDro), LILACs and ScienceDirect. Medical Subjects Headings (MeSH) descriptors and other keywords combined with Boolean operators were used. The terms were: “spasticity”, “muscle spasticity”, “electromagnetic stimulation”, “electromagnetic therapy”, “pulsed electromagnetics”, “electromagnetics fields” and “magnetic field therapies”.

The search was filtered to full-text clinical trials papers. No date and language filters were applied. Table 1 shows the different search combinations.

The PICOS (Population, Intervention, Comparison, Outcomes and Study design) model [28] was used to establish the inclusion criteria: (I) Population: humans with spasticity; (II) Intervention: Treatment with electromagnetic fields administered to the lower or upper limbs or spine; (III) Comparison: placebo, no treatment, a different electrotherapy modality or any other intervention; (IV) Outcomes: related to spasticity; (V) Study design: controlled clinical trials. Articles where participants were people with spasticity, but the outcome data were not provided or those where transcranial therapy was used, were excluded.

### 2.2. Study Selection Process and Data Extraction

First, a search was carried out by combining the keywords in the different databases. Duplicate articles were then removed using the Rayyan tool (https://www.rayyan.ai/, accessed on 27 February 2022). Subsequently, studies were selected or excluded. Two reviewers (M.J.V.-G. and G.G.-M.) carried out the process of study selection, review and systematic data extraction. A third reviewer (R.M.-V.) was involved in reaching consensus in case of controversy.

The following information was extracted from each article included in the review: authors, type of intervention, disease or pathology causing spasticity, number of subjects, frequency of sessions per week, time of each session, total duration of the intervention, outcome measures, measurement instrument, device used for the application of magnetic fields, parameters used and results obtained.

### 2.3. Risk of Bias and Assessment of the Methodological Quality of the Included Studies

The risk of bias was calculated for each selected study using the Cochrane Collaboration tool [29]. The following types of bias were assessed: selection bias, performance bias, detection bias, attrition bias, reporting bias and other biases.

In order to assess the quality of the articles used for the systematic review, the PEDro scale [30] was used. This scale consists of 10 items: randomization, concealed allocation, comparability at baseline, blinding of subjects, blinding of therapists, blinding of assessors, more than 85% follow-up for at least one key outcome, intention-to-treat analysis, statistical comparison between groups, and point and variability measures for at least one key outcome. Items are scored as yes (1) or no (0), and the maximum score is 10 points. An additional criterion (item 1: selection criteria) that relates to external validity (applicability of the test) is included to complete the Delphi list, but this criterion is not used for the calculation of the scale score [31]. Taking into account the established criteria, a study with a PEDro score of 6 or higher is considered as evidence level 1 (6–8: good; 9–10: excellent), and a study with a score of 5 or lower is considered as evidence level 2 (4–5: acceptable; <4: poor).

## 3. Results

### 3.1. Selection of Studies

Once the database searches were completed, by combining the different key words, a total of 521 documents were obtained, of which 10 studies were finally included in the systematic review [32,33,34,35,36,37,38,39,40,41]. Figure 1 shows the flow chart of the search process.

A meta-analysis was attempted with the EPIDAT program of the four studies [32,33,40,41] that provided numerical data for its performance; however, given the methodological, clinical and statistical heterogeneity, it was not possible.

### 3.2. Data Extraction

#### 3.2.1. Characteristics of the Subjects

There were a total of 460 participants with an age ranging from 32 [38] to 76 years [40], out of which 40% were men.

The largest sample studied was that of Lappin et al. [34] with a total of 117 subjects and the smallest was that of Serag et al. [37] with 26. In 70% of the studies the sample was between 26 and 38 years of age [32,33,35,36,37,39,40] and 63% were women.

Regarding the pathology that had originated the spasticity, 50% of the studies were multiple sclerosis [32,33,34,35,37], although in the study by Krause et al. other spinal diseases were also included [35] and the other 50% of the trials dealt with stroke [36,38,39,40,41], although in the article by Krewer, subjects with traumatic brain injury were also included [38].

#### 3.2.2. Main Characteristics of the Studies

Table 2 shows the main characteristics of the interventions performed in the different studies that make up the present review. We can highlight that 80% of the trials used sham stimulation in the control group [32,33,34,35,36,37,38,39] and in the other 20% in the intervention group RPMS was applied in the antagonists and agonists and in the intervention group only in the antagonists. In two of the trials there was a control group with healthy individuals [35,39]. Concerning the way the electromagnetic field was falsely applied, sometimes the device was used switched on but without applying any intensity [32,33,34,35,37,38,39] or it was applied without intensity and in another location [36].

In 3 articles [36,38,40], both the intervention and control groups also had a complementary physiotherapy program.

In 3 of the 10 trials the electromagnetic fields were administered in the form of PEMP [33,34,36] and in the other 7 the therapy was RPMS [32,35,37,38,39,40,41].

There was much heterogeneity in terms of the device used, in two of the articles it was portable [33,34]. The oldest trial (1997) used an oil-cooled coil [32] and the most recent [40,41] (2018 and 2022) used an inductive system. There were articles in which the brand of the device was mentioned: Enermed [33,34], Magstim Rapid [35,39], BTL-6000 [40], Dantec-Maglite [37], P-Stim 160 b [38].

Some were placed on the spine [32,35,36,37] and others on the upper limbs [38,39,40,41]. In one trial, the device was placed on an empirically determined acupuncture point on the spine, shoulder or hip [33].

Some articles specified that they were placed directly on the skin [33,34] and in other cases, there was no contact [35,37,40].

In terms of frequency, the most commonly used frequency was 25 Hz [32,34,38]. In addition, 1 Hz [37], 4–13 Hz [33], 20 Hz [35], 50 Hz [39] and 25–150 Hz [40] were used.

The time of application ranged from 8 min [39,40] to 24 h a day [33,34], although 9 min [41], 20 min [36,38] or 25 min were also applied [32].

The intensity was gradually increased up to 0.7 Tesla [32], increased by around 20% for the stimulation series [35], was increased or decreased by patient’s tolerance [40] or was set at 10% above the level that evoked a movement [38].

The sessions were performed once a day but sometimes twice a day [32,38] or during 24 h [33] and the total number ranged from 1 session [39] to 10 sessions [40]. Concerning the treatment time it was very variable: 1 day [39], 7 days [32], 10 days [40], 14 days [37,38], 56 days [33] or 70 days [34].

As for how to assess spasticity, the Asworth scale [32,35,37,40,41] was most commonly used, followed by electromyographic parameters [32,36,39]. Among them, the Hmax/Mmax ratio [36] was recorded. The Watenberg’s pendulum test [35] and other scales such as the Modified Tardie Scale [38] were also used.

There were also self-reported questionnaires on difficulties encountered in activities of daily live [32] or on performance [33] due to spasticity and a self-reported spasm frequency [37]. The Barthel Index [40,41] and other more specific multiple sclerosis scales such as the Expanded Disability Status Scale (EDSS) [33] and the Multiple Sclerosis Quality of Life Inventory (MSQLI) [34] were also used for the same purpose.

Other variables measured were upper limb functionality measured by the Fugl-Meyer Assessment [38] and ankle functionality through passive and active range of motion measured with a goniometer, sometric muscle strength and resistance of plantar flexors to stretch with a dynamometer [39] or walking speed through the 25 feet walking test [37].

In addition, a quantitative electroencephalografic during a language task [33] and a TMS were used to see the ipsilateral cortical motor representation [39].

Regarding the results obtained in terms of spasticity, in 80% of the studies were positive. Improvements were found in stretch reflex threshold [32,35,39], self questionnaire about difficulties related to spasticity [32], clinical spasticity score [32], performance scale [33], Ashworth scale [32,35,37,40,41], spastic tone [35], Hmax/Mmax Ratio [36] and active and passive dorsal flexion [39].

Only 2 articles found mixed results [34] or a limited effect [38] for spasticity.

Regarding other measured variables, there was improvement in Barthel index [40,41], fatigue [33,34], quality of life [34] and sensory function [38] but not gait speed [37].

### 3.3. Methodological Quality Assessment

80% of the studies were of good quality [32,33,34,36,37,38,39,41] and the other 20% [35,40] were of acceptable quality. Table 3 shows the score for each study.

### 3.4. Risk of Bias Assessment

The results of the risk of bias can be observed in Figure 2. It should be noted that the risk of bias is low in relation with selection bias referring to random sequence generation as in only one article it was not fulfilled [35] and another had uncertain risk [33]. Only one article specified allocation concealment [37]. 80% of the articles had a low risk of bias in relation with performance bias [32,33,34,35,36,37,38,39]. With respect to reporting bias all of them were unclear risk because none of the articles specified whether they had registered the clinical trial in a database before (Figure 3).

## 4. Discussion

A systematic review has been carried out to synthesize the scientific evidence regarding the use of electromagnetic fields, applied peripherally, in the treatment of the spasticity.

With regard to the characteristics of the sample, it was homogeneous in terms of the pathologies studied, as half were in sclerosis and the other half in stroke. It should be borne in mind that, although the incidence and prevalence figures for spasticity vary [42], multiple sclerosis and stroke are two of the most prevalent pathologies, with an estimated 80% [43] and 42.6% [44], respectively.

It would be advisable to study the effect of magnetic fields in other diseases where prevalence is also high, such as spinal cord injuries where it is estimated that between 40 and 78% have spasticity [45] or in cerebral palsy where the percentage is even higher (72–91%) [46].

Studies have been carried out on rats with induced spinal cord injuries with good results demonstrating the viability and efficacy of this therapeutic strategy for spasticity; however their results have not yet been demonstrated in humans [47].

Regarding the type of electromagnetic fields used in the treatment of spasticity, the most investigated therapy has been transcranial [17,18,19,20,21,22]. In our extensive literature search on peripherally delivered electromagnetic fields, only three of the articles [33,34,36] studied the efficacy of PEMF in spasticity. This type of therapy has been most studied in osteogenesis stimulation [48] and in muscleskeleton disorders such as osteoarthritis [49], fibromyalgia [50], rotator cuff tendinitis [51] and lateral epycondilitis [52].In neurology its use has focused on diabetic peripheral neuropathy [53]. The rationale for this therapy is that it has a stimulating effect on biological processes [54].

The other 7 articles [32,35,37,38,39,40,41] used RPMS. This system activates peripheral nerves and muscles without stimulating skin nociceptors while limiting pain [12].

When compared to other types of electrotherapy that have been shown to be effective in improving spasticity such as neuromuscular electrical stimulation NMES [55,56], RPMS-induced pain is significantly less than NMES-induced pain, even when using the same stimulation intensity [57]. Therefore, RPMS simultaneously provides stronger stimulation than NMES and limits pain [58]. A review of non-pharmacological interventions used for the treatment of spasticity in people with multiple sclerosis, there was no evidence for the use of NMES. Furthermore, its depth of stimulation is very shallow and it has some adverse effects including skin burns, dermatitis and pain [59]. However the authors found that magnetic stimulation and electromagnetic therapies were beneficial although with a ‘low level’ of evidence [60].

Therefore, an important aspect of this type of therapy is that there are no known adverse effects; only one of the articles mentions [32] that magnetic stimulation evoked contraction of the mid-thoracic paravertebral and intercostal muscles, causing a sensation of tension around the chest but without cardiac involvement, recommending for future studies a more careful placement of the magnetic coil.

With respect to the number and duration of sessions, device used, doses, intensity and place of therapy administration, there is no clear pattern. All these parameters have been inhomogeneous.

Concerning the measuring instruments, they were very heterogeneous, but it must be taken into account that the term “spasticity” is multifactorial, which makes it difficult to evaluate, and there are different measurement methods that can be divided into non-instrumental and instrumental methods, based on neurophysiological studies of spinal reflexes [61]. In the articles of our review, the most commonly used method was the modified Ashowrth scale [32,35,37,40,41], which is in agreement with the scientific literature, as it does not require any tools and is easy to apply [42], but it is not sensitive enough for measuring the characteristics that distinguish spasticity from other tone alterations [62]. The Tardie scale is considered a better option as it compares muscle reaction to passive movement at different speeds [63], although it was only used in one of the articles in our review to measure spasticity in stroke [38]; however it is most reliable in cerebral palsy [64]. Other indirect clinical assessment methods would be those aimed at measuring the impact of spasticity on the individual. The articles in our review used the Fugl-Meyer scale to measure limb functionality [38], the Barthel Index [40,41] to assess functionality in daily living, and gait scales such as the 25 feet walking test [33] to assess walking speed. There were also more specific multiple sclerosis questionnaires such as the Expanded Disability Status Scale (EDSS) [33] and the Multiple Sclerosis Quality of Life Inventory (MSQLI) [34].

As for quantitative or instrumental methods, the most accurate would be electromyography and the pendulum test [42]. In our study, only two studies used these more objective methods, in one of the articles the pendulum test [36] and the other the Hmax/Mmax ratio [35]. Given the wide heterogeneity in the measurement of spasticity, it would be advisable to be more specific in its measurement, using more objective instrumental methods that could be supported by more specific, reliable and validated non-instrumental methods according to the pathology studied. Gómez-Soriano et al. recommended a combination of the different assessment tools such as the scales, neurophysiological measures, biomechanical methods to know the degree of spasticity present in the patient [6].

Regarding the results, in most of the articles there is an improvement of spasticity, in agreement with what has been found in the scientific literature where there have been trials in rats [47], observational studies [65] or clinical case studies [66,67,68]. What is not clear are the mechanisms of action and the maintenance time of the antispasticity effect. In relation to the mechanisms of action, it would be useful to acquire more knowledge in order to be able to be more specific in the treatment [48]. According to the maintenance time of the anti-spasticity effect, it is stated that it did not outlast 24 h; however, in the articles reviewed in this study, this effect lasted longer, up to 30 days [41].

The present study has several strengths, including the broad and easily reproducible search strategy applied to nine major medical databases. In addition, studies have a good or acceptable quality. It should be noted that in physiotherapy studies it is difficult to blind subjects but with this type of therapy this has been achieved in most studies.

However, some limitations need to be addressed before drawing conclusions from the results of the present analysis. Despite the extensive literature search was carried out, with no date limit, only a few articles were found. The first one in 1996 and the last one in 2021. The scientific evidence of electromagnetic fields on spasticity has been written about for more than two decades, and there are few randomized clinical trials.

Another limitation was related to the great heterogeneity among the different studies. It was so extensive that a meta-analysis could not be performed. There was little uniformity in device used, time of application, duration, frequency, intensity and how and where it was applied. Furthermore, different pathologies and types of electromagnetic fields applied in the periphery were analyzed. In addition, the measurement instruments were not very objective and very heterogeneous.

Large, multi-center, double-blind, controlled studies are needed to draw conclusions for the therapeutic management of spastic patients, as well as comparative studies of treatment protocols with standardized methodology should be carried out.

## 5. Conclusions

Based on the studies included in this review, it appears that the peripheral application of electromagnetic fields is beneficial in spasticity. Improvements have been found in stretch reflex threshold, self-questionnaire about difficulties related to spasticity, clinical spasticity score, performance scale, Ashworth scale, spastic tone, Hmax/Mmax Ratio and active and passive dorsal flexion. However, results must be taken with caution due the small number of articles and to the large heterogeneity in terms of the device used, application site, treatment time, intensity, number of sessions and duration of therapy. The most commonly used form of application was RPMS and the frequency was 25 Hz. In future studies, it would be interesting to define and agree on the parameters to be used, as well as the way of assessing spasticity, in order to be able to make a more objective comparison of its efficacy compared to other therapeutic alternatives.

## Figures and Tables

**Figure 1 jcm-11-03739-f001:**
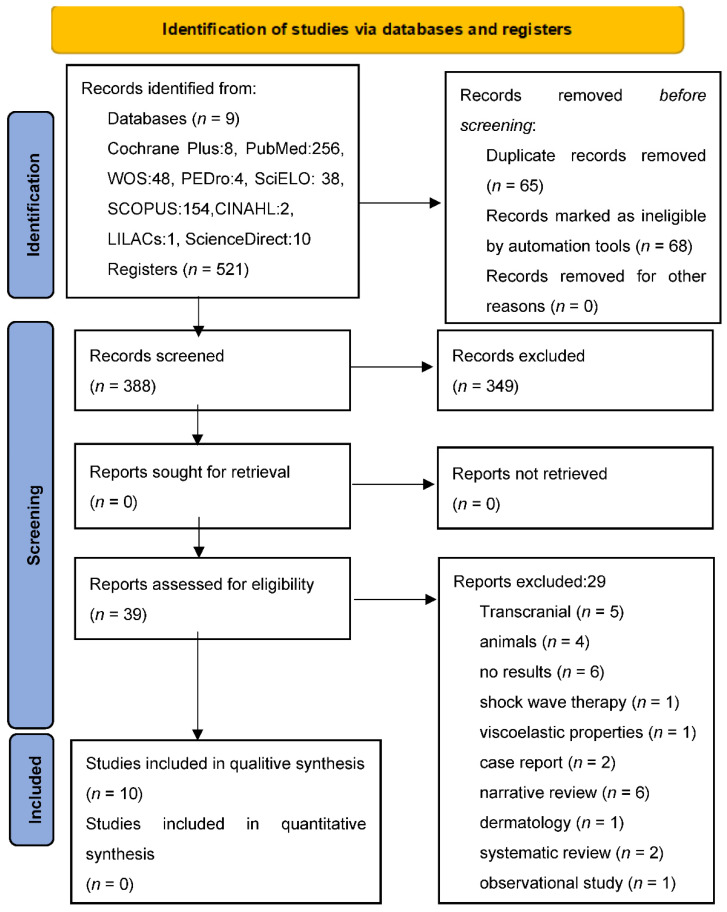
PRISMA 2020 flow diagram. From: Page MJ, McKenzie JE, Bossuyt PM, Boutron I, Hoffmann TC, Mulrow CD, et al. The PRISMA 2020 statement: an updated guideline for reporting systematic reviews. BMJ 2021;372: n 71. doi: 10.1136/bmj.n71.

**Figure 2 jcm-11-03739-f002:**
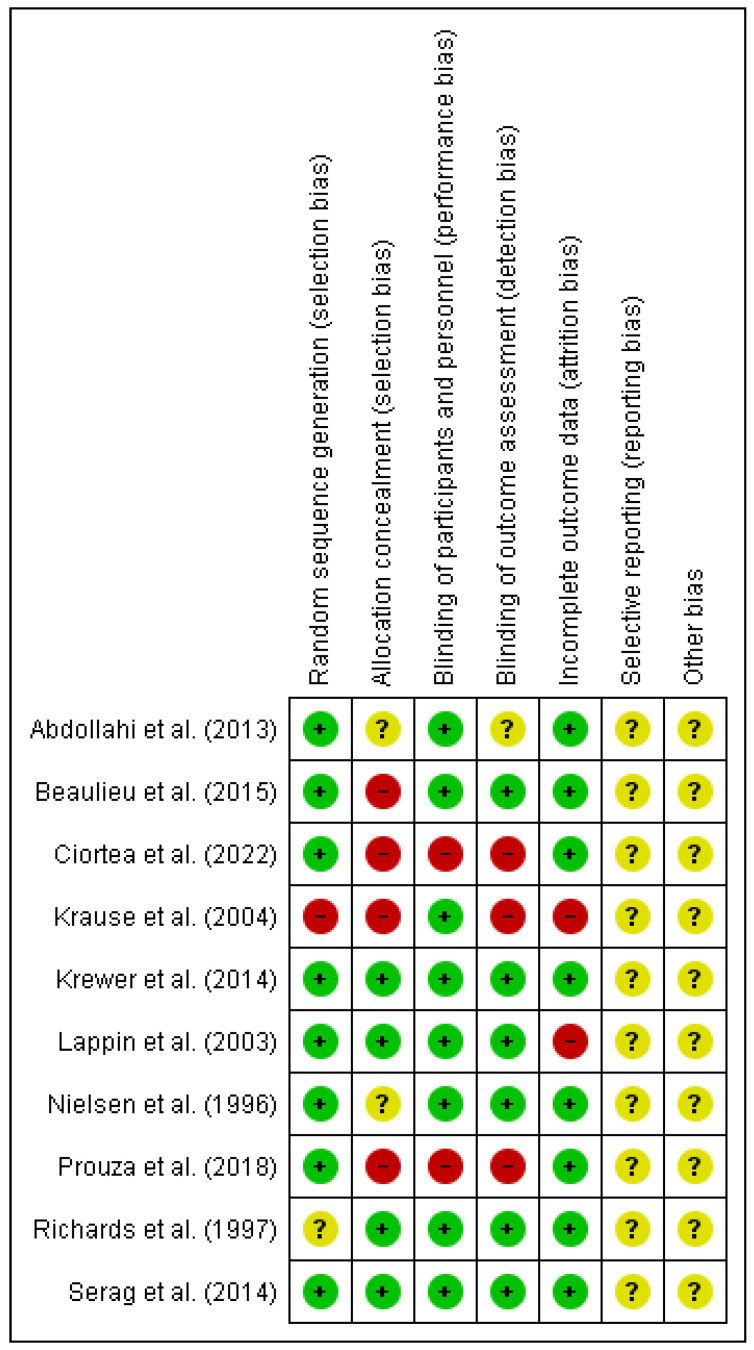
Risk of bias summary [32,33,34,35,36,37,38,39,40,41].

**Figure 3 jcm-11-03739-f003:**
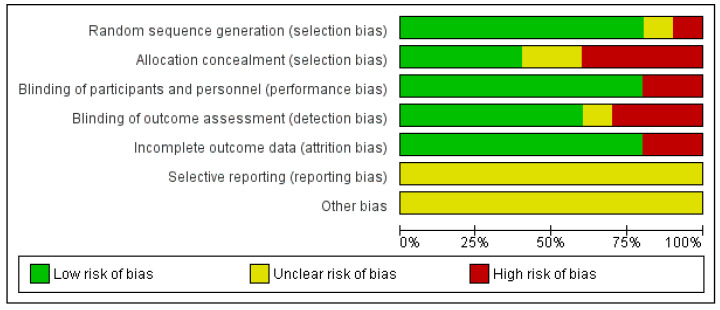
Risk of bias graph.

**Table 1 jcm-11-03739-t001:** Search combinations.

Databases	Search Strategy
Cochrane Plus	(field electromagnetic) AND spasticity in title abstract keyword
PubMed	(muscle spasticity OR musc* tone OR spastic*OR (musc* stiffness) AND (electromagnetic OR pulsed electromagnetic*OR c OR electromagnetic field* OR electromagnetic radiation OR magnetic field therapy)
WOS	TITLE-ABS-KEY (((musc* AND tone) OR spastic* OR (musc* AND stiffness)) AND (eletromagnetic AND field*) OR (electro-magnetic AND therapy) OR (pulsed AND electromagnetics) OR (electromagnetics AND fields) OR (electromagnetic AND wave) OR (magnetic AND field AND therapies)) AND (LIMIT-TO (DOCTYPE, “ar”)) AND (LIMIT-TO (EXACTKEYWORD, “Human”) OR LIMIT-TO (EXACTKEYWORD, “Humans”))
PEDro	spasticity AND electromagnetic fields
SciELO	Electromagnetic field in title
SCOPUS	((Musc* AND Tone) OR spastic* OR (musc* and stiffness)) AND (eletromagnetic field*) OR (Electro-magnetic therapy) OR (pulsed electromagnetics) OR (electromagnetics fields) OR (electromagnetic wave) OR (magnetic field therapies)
CINAHL	AB electromagnetic fields AND AB spastic*
LILACs	(electromagnetic field) AND spast* in title, abstract, subject
ScienceDirect	spasticity AND (electromagnetic fields) NOT transcranial. Research articles. Subject area: Nursing and Health Professions

**Table 2 jcm-11-03739-t002:** Principal studies characteristics.

Authors (Year)/Design	Study Groups/Mean (SD)/Gender	Measuring (Evaluation Instruments)	Intervention	Parameters/Device Used	Results
Nielsen et al. [32]/1996 Multiple sclerosis	N = 38 M = 44 26 women 12 men IG: 21 CG: 17	- Self-Questionnaire: daily day activities with one score only (0–10). Focus on the particular difficulties related to spasticity. - spasticity: Ashworth scale/EMG - reflex activity:conventional clinical grading. Evaluation: before,1st day, 8th day, 16th day.	IG: RPMS CG: sham stimulation twice daily for 7 consecutive days	RPMS: biphasic waveform with a pulse width of 240 µsec, a rise time of 60 µsec, and a maximum magnetic field of 1.2 Tesla, with repeated periods of stimulation for 8 s at 25 Hz followed by 22 s of repose, 25 min. - Relaxed supine position. - Intensity: was gradually increased to 0.7 Tesla within a few minutes Device: magnetic stimulator with an oil-cooled coil. The coil has an outer winding diameter of 13.4 cm consisting of a 16-turns copper tube.	Spasticity: IG improvement 18% for the clinical score and 27% for the stretch reflex threshold, 78% (14/18) of the treated patients improved clinically and 50% (9/18) improved their stretch reflex threshold. - Self questionnaire: IG: improved 22% (*p* = 0.007) vs. CG 29% (*p* = 0.004). - Clinical spasticity score improved −3.3± 4.7 vs. CG and 0.7± 2.5 (*p* = 0.003). - Stretch reflex threshold increased IG: 4.3+ 7.5 vs. CG −3.8 ± 9.7 (*p* = 0.001).
Richards et al. [33]/1997/RCT Multiple sclerosis	N = 30 IG: 15 CG: 15	- Clinical rating (EDSS). - Patient-reported performance scales. - Quantitative electroencephalography during a language task.	IG: PEMP CG: sham stimulation Between 10 and 24 h a day, 2 months	PEMP: frequency 4–13 Hz range (50–100 milliGauss) Device: magnetic pulsing device (Enermed)	Improvement performance scale improved combined rating for bladder control, cognitive function, fatigue level, mobility, spasticity, and vision (IG −3.83 ± 1.08, *p* < 0.005; CG −0.17 ± 1.07). - spasticity (functional scales disability) average change IG −0.80 (0.23) (*p* < 0.005) vs. CG −0.17 (0.24)
Lappin et al. [34]/2003/cross-over Multiple sclerosis	N = 117 (41–62) IG: 117 89 women 28 men	- spasticity: (MSQLI) - fatigue: (MSQLI) - bladder control: (MSQLI) - quality of live: (MSQLI) Evaluation: after each of the 2 treatment sessions.	IG: PEMP:2 weeks IG: Sham stimulation: 2 weeks 10 weeks, with 2, 3 week treatment sessions separated by a 2-week washout period.	- PEMP: pulsed electromagnetic signals 1 to 25 times per second. Duration of each pulse: 1 milisecond, input wave form is a square wave. 24 h/day. - Over the brachial plexus. - Device: Enermed, Energy Medicine Developmentes, Inc., Vancouver, British Columbia.	- Improvements in fatigue, quality of life on the active device. - Mixed results for spasticity: Enermed ss 0.24 (0.79) vs. Sham ss 0.13 (0.69) (*p* = 0.04), not difference in the treatment effects on daily diary - muscle spasm/spasticity measured using the MSQLI at the end of each sessions: statistically significant differences (*p* = 0.04).
Krause et al. [35]/2004/RCT Spinal diseases (multiple sclerosis, familial spastic spinal paralysis, transverse mielitis, spinal vasculitis)	N = 31 IG: 15 spinal lesions (M: 34.2) CG: 16 healthy subjects (M: 42.3)	Spasticity: MAS/Wartenberg’s pendulum test. Evaluation: 2, 4, 24, 48 h CG: only 3 times	IG: RPMS CG: sham stimulation	- RPMS unilateral stimulation nerve roots L3/L4 of the more spastic leg. - RPMS each series of stimulations was applied 10 times, each series of stimulations lasting 10 s at a frequency of 20 Hz. The interstimulus was 4 s. Altogether, 2000 single magnetic stimuli were given on more affected leg. - Subjects seated. - The intensity of the motor threshold was increased by around 20% for the stimulation series. - Device: Magstim Rapid with a maximum output of 1275 T with circular coil with a diameter of 90 mm positioned at the level of vertebrae L3/L4.	- Ashworth scale a peak reduction 4–24 h after stimulation (ipsilateral and contralateral, *p* < 0.008). Contralateral side also decreased. - velocity of the first swing of the lower leg increases in both legs (ipsilateral: 3620 s^−1^ before to 4280 s^−1^ after 24 h, *p* < 0.008; contralateral: 3440 s^−1^ before to 4240 s^−1^ after 24 h, *p* < 0.008 - Intensity for determining the motor threshold higher in IG tan in CG (43% of the maximal stimulator output compared with 32%, *p* = 0.01 - Spastic tone decrease seen as an increase in swing velocity of the lower limbs (ipsilateral and contralateral). The reduction of spastic tone tended to be more pronounced contralaterally, lasted for around 20 h. - IG motor threshold for the paraspinal magnetic stimulation higher tan CG. The spastic tone decreased between 4 and 24 h after stimulation. This effect was slightly more pronounced in the contralateral extremity. Furthermore, the stimulation motor threshold of the patients was significantly raised. RPMS decreases spasticity for 1 day not only on the ipsilateral but also on the contralateral side.
Abdollahi et al. [36] *2013/RCT* *Stroke*	N = 30 (50) IG: n = 10 Sham G: n = 10 CG: n = 10	Spasticity and alpha motoneuron excitability: (Hmax/Mmax Ratio)	IG: PEMP+PT Sham G: Sham+PT CG: PT PT: programm of lower limb (warm up for 10 min, functional electrical stimulation for dorsi flexor muscles 20 min, mat exercise to hypertonecity inhibition, stepping and weight bearing on the affected side 20 min, treadmill walking 10 min).	IG: PEMP 20 min in position lying on the spinal cord	Hmax/Mmax Ratio decreased in IG, Sham G, CG after treatment but more in IG (*p* = 0.012).
Serag et al. [37] 2014/RCT Multiple sclerosis	N = 26 IG = 18 34.6 ± 9.2 CG = 8 32 ± 11.2	- Spasticity (MAS) - self-reported spasm frequency. - Degree of pain - walking speed:25 feet walking test. Evaluation: before, 2nd/4th weeks.	IG: active RPMS CG: sham RPMS - On alternate days, 2 weeks. 6 ss	IG: 1 Hz, RPMS at a fixed intensity of 45% applied bilaterally at L2-4 spinal roots, 2 cm from midline. Stimulation:Dantec-Maglite magnetic stimulator with a figure of eight coil.	- IG: Improved muscle spasticity (MAS) (*p* = 0.05) and spasm frequency and intensity (*p* < 0.0001). - IG/CG:No difference in duration to complete 25 feet test or body pain
Krewer et al. [38]/2014 Stroke/traumatic brain injury	N = 66 IG = 31 55 ± 13 12 women 19 men CG: 32 54 ± 13 13 women 19 men	Spasticity: Modified Tardieu Scale/Fugl-Meyer Assessment (arm score) Evaluation: before, 2nd treatment/4th weeks. Tardie scale: 3 ss	IG: active RPMS + PT CG: sham RPMS + PT 2 weeks, 2 times a day PT: self-administered ROM exercises/slow passive stretches executed 30 s and proprioceptive neuromuscular facilitation movement 30 s.	- RPMS: 5000 - stimuli at a frequency 25 Hz, a train duration of 1 s/ intertrain interval of 2 s. - Intensity was set at 10% above the level that evoked a wrist or elbow movement taken at rest. Stimuli were distributed consistently among extensors and flexors of the upper and lower arm. 20 min. - CG: 20 min sham stimulation - Device: Signal software (Signal for Windowsa), and the digital outputs were fed through an analogue-digital converter (Micro 1401 mk IIa) into the magnetic stimulator (P-Stim 160b): generated double cosine pulses with a magnetic induction of maximally 1 tesla.	- Limited effect on Spasticity (Tardieu > 0) was present in 83% of wrist flexors, 62% of elbow flexors, 44% of elbow extensors, and 10% of wrist extensors. - G vs. CG: short-term effects on spasticity for wrist flexors (*p* < 0.048), and long-term effects for elbow extensors (*p* < 0.045). Limited effect on - No effect on motor function. Arm motor function IG Med: 5 vs. CG Med: 4. - Effect on sensory function
Beaulieu et al. [39] 2015/RCT Stroke	N = 32 IG: 9 (51 ± 15) 5 women 4 men Sham C: 9 (55 ± 11) 6 women 3 men CG: 14 healthy subjects (50 ± 7) 8 women 6 men	- Ankle impairments on the paretic side (EMG recordings, ROM, active and passive with goniometer, sometric muscle strength and resistance of plantar flexors to stretch(dynamometer) - Ipsilateral TA cortical motor representation (TMS)	IG: RPMS over the paretic TA - 1 session lasting 2–3 h including rest breaks	IG: RPMS, biphasic waveform, 400-ms pulse width, rapid-rate magnetic stimulator Rapid2 Magstim) were delivered at a theta-burst frequency, i.e., 5-Hz bursts of three 50-Hz pulses each, Intermittent theta-burst stimulation of 2 s ON 8 s OFF (600 pulses) was applied for 190 s. Using an air film cooled figure-of-eight coil. The coil was held tangentially on the skin overlying the paretic TA muscle. Intensity was set at 42% of the maximal stimulator output. - Sham G/CG: = parameters with low intensity (5% of maximal stimulator output) with the coil positioned directly above the metatarsals	IG: ankle dorsiflexion mobility and maximal isometric strength increased and resistance to plantar flexor stretch decreased. - Sham stimulation yielded no effect. - A significant group time interaction was detected for plantar flexor resistance to stretch (F55.71; P50.03) and a trend only for active DF ROM (F53.92; P50.065). Planned comparisons determined that after RPMS plantar flexor resistance to stretch was reduced (mean decrease of 2.4 + 2.0 kg; P50.0007) with concomitant increases in active DF ROM (mean increase of 7.8 + 7.3u; P50.0005), passive DF ROM (mean increase of 2.2 + 1.9u; P50.03) and DF strength (mean increase of 1.32 + 1.25 kg; P50.05)
Prouza et al. [40]. 2018/RCT stroke	N = 30; 66.93 ± 9.31 25 women 5 men IG: 30 CG: 30	- Spasticity (MAS) - Activities daily living: Barthel Score	IG: RPMS on agonist and antagonist + PT IC: RPMS on antagonist + PT PT: Bobath approach, proprioceptive neuromuscular facilitation (Kabat)	IG: RPMS; 10 ss, 9 min, frequency 25–150 Hz pulsed duration 280 microseconds, daily, above the pathological area (contactless delivery), firstly, agonist muscle in the upper extremities was stimulated to achieve post-facilitatory inhibition; subsequently, the weakened antagonist muscles were stimulated. The intensity of the therapy was set at the beginning and was increased/ decreased by patient’s tolerance CG: 10 ss, 8 min 50–100 Hz Pulse duration 0.2–2.0 microseconds, 10 daily therapies on the antagonist muscles of the upper extremities Device: BTL-6000 Super Inductive System, BTL Industries Ltd.).	- MAS: IG improved results up to 66% decreasing spasticity from 2.33 ± 0.90 in the beginning to 0.87 ± 0.64 points vs. CG improved up to 31% decreasing spasticity from 2.13 ± 0.74 in the beginning to 1.47 ± 0.74 points (1-month follow-up) - Barthel Index, IG, 81% level of improvement vs. CG 72% level of improvement (1-month follow-up).
Ciortea et al. [41] 2022/RCT Stroke	N = 60 (62) IG: 29 15 women 14 men CG: 31 15 women 16 men	- Upper extremity functional index: (MAS) - Activities daily living: Barthel Score Evaluation: Before/10th day/30th day	IG: RPMS agonists+antagonists muscles + PT CG: RPMS antagonists muscles + PT 10 ss	IG: RPMS, Super inductive system, 10 ss 9 min. On the agonist muscles (flexors forearm), 1 min + antagonist (extensors forearm) 8 min + PT CG: RPMS, Super inductive system, antagonist (extensors forearm) 8 min + PT	- MAS increased - 10th–30th: IG (−0.28 ± 0.53, *p* = 0.001 vs. CG (−0.52 ± 0.51, *p* < 0.001. - Barthel increased 1st–30th: IG/CG (−1.93 ± 1.60, *p* < 0.001 vs. −1.87 ± 1.09, *p* < 0.001) and decreased 10th–30th IG/CG (0.35 ± 0.94, *p* = 0.064 vs. 0.55 ± 0.96, *p* = 0.005. - % participants improved MAS in GI 100% vs. GC 67,%, *p* = 0.004.

PEMP: pulsed electromagnetic field; RPMS: Repetitive peripheral magnetic stimulation; IG: experimental group; Sham G: sham group; CG: control group; PT: physiotherapy; min: minutes; h: hours; Hmax/Mmax Ratio: Hmax-to-Mmax ratio, electromyographic ratio; ss: sessions; MAS: Modified Ashworth Scale; NRS: numerical rating scale; ROM: active range of motion; FMA-UE: Fugl Meyer Assessment scale (subscale A; shoulder/elbow/forearm, B; wrist, C; hand, D; coordination/speed); M: mean; SD: deviation standard; TA: tibialis anterior; TMS: transcranial magnetic stimulation; RCT: randomized clinical trials; ROM: range of movement; DF: dorsal flexion; EMG: electromyogram; MSQLI: Multiple Sclerosis Quality of Life Inventory; EDSS: Expanded Disability Status Scale.

**Table 3 jcm-11-03739-t003:** Methodological quality assessment (PEDro Scale).

Criteria	Nielsen et al. [32]	Richards et al. [33]	Lappin et al. [34]	Krause et al. [35]	Abdollahi et al. [36]	Serag et al. [37]	Krewer et al. [38]	Beaulieu et al. [39]	Prouza et al. [40]	Ciortea et al. [41]
Eligibility criteria	Y	Y	Y	N	N	Y	Y	Y	Y	Y
Randomization	Y	Y	Y	N	Y	Y	Y	Y	Y	Y
Allocation concealed	N	N	N	N	Y	Y	Y	Y	N	N
Baseline comparability	Y	Y	Y	N	N	Y	Y	Y	Y	Y
Subject blinding	Y	Y	Y	Y	Y	Y	Y	Y	N	N
Therapist blinding	N	N	N	N	N	N	N	N	N	N
Evaluator blinding	Y	Y	Y	N	N	Y	Y	Y	N	N
Appropriate continuation	Y	Y	Y	Y	Y	Y	Y	Y	Y	Y
Intention to treat	Y	Y	N	Y	Y	Y	Y	Y	Y	Y
Comparison between groups	N	Y	Y	Y	Y	Y	Y	Y	Y	Y
Specific measurements and variability	Y	Y	Y	Y	N	N	Y	Y	Y	Y
Total PEDro Score	7	8	7	5	6	8	8	8	8	6

“N” indicates those items that not scoring; “Y“ indicates those items score.

## Data Availability

Not applicable.

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
