# Peer review of "Effects of Peripheral Electromagnetic Fields on Spasticity: A Systematic Review"

_jcm, 2022, doi:10.3390/jcm11133739_

Round 1

Reviewer 1 Report

The interaction of a magnetic field in different biological systems has been evaluated by different researchers in this area of knowledge for several decades. In this review you focus on the effect of an electromagnetic field in stimulating stiff and tense muscles.

However, some details of format and content have to be adjusted:

A. Although the search technique for a review is very important, however according to the title “Effects of peripheral electromagnetic fields on spasticity. A systematic review”, should focus a little more in depth on this medical topic in some sections:

ABSTRACT: Mentioning the search technique and the databases in the abstract I think is not the focus. It should focus on the effects of electromagnetic fields, frequency and amplitude of a stimulation with these characteristics.

METHODS (Search strategy): In this section you should include everything related to the search process, the use of different search engines, databases, phrases used, search diagram, etc. And do not place in another section that downplays the central theme of the scientific review.

CONCLUSION: You should be more conclusive in this section. I believe that the conclusion for a scientific review should be strengthened. For example, it concludes on the secondary effects, on the benefits, on the amplitude and frequency of stimulation, among other parameters referring to stimulation in biological systems.

B. The references are not in the format indicated by the journal:

You can check the proper format in the instructions for authors section

(https://www.mdpi.com/journal/jcm/instructions)

For example:

FOR AN ARTICLE:

Author 1, A.B.; Author 2, C.D. Title of the article. Abbreviated Journal Name Year, Volume, page range.

FOR A BOOK:

Author 1, A.; Author 2, B. Book Title, 3rd ed.; Publisher: Publisher Location, Country, Year; pp. 154–196.

FOR A BOOK CHAPTER:

Author 1, A.; Author 2, B. Title of the chapter. In Book Title, 2nd ed.; Editor 1, A., Editor 2, B., Eds.; Publisher: Publisher Location, Country, Year; Volume 3, pp. 154–196.

Author Response

ITEMIZED LIST OF THE REVIEWERS COMMENTS

Manuscript ID:   jcm-1746312

Title: “EFFECTS OF PERIPHERAL ELECTROMAGNETIC FIELDS ON SPASTICITY. A SYSTEMATIC REVIEW”

Dear Reviewer,

We greatly appreciate the editor´s and reviewers’ kind and encouraging comments about our study. We have followed their suggestions, trying to incorporate them into the revised version of our manuscript. We uploaded the tracked changes manuscript, the clean version revised manuscript and itemized point-by-point response to the reviewer’ comments are presented below.

Editor´s and Reviewers´ comments:

*Reviewer 1

RV: Reviewer

AA: Authors

Response to reviewer 1 Coments.

  1. Although the search technique for a review is very important, however according to the title “Effects of peripheral electromagnetic fields on spasticity. A systematic review”, should focus a little more in depth on this medical topic in some sections:

  1. Point 1. ABSTRACT: Mentioning the search technique and the databases in the abstract I think is not the focus. It should focus on the effects of electromagnetic fields, frequency and amplitude of a stimulation with these characteristics.

 Response 1. Changes suggested by the reviewer in the summary have been made to give more importance to parameters that were not mentioned before. Thank you for helping us with your input.

  1. Point 2. METHODS (Search strategy): In this section you should include everything related to the search process, the use of different search engines, databases, phrases used, search diagram, etc. And do not place in another section that downplays the central theme of the scientific review.
  2. Response 2. As requested by reviewer 1, two of the material and methods sections have been merged so that everything related to the search process is in the search strategy section (pages 2 and 3, lines 73-99). Thank you very much for make better our manuscript.

  1. Point 3. CONCLUSION: You should be more conclusive in this section. I believe that the conclusion for a scientific review should be strengthened. For example, it concludes on the secondary effects, on the benefits, on the amplitude and frequency of stimulation, among other parameters referring to stimulation in biological systems.
  2. Response 3. In order to improve the quality of our manuscript, and in an attempt to answer the 3 recommendation suggested by the reviewer, conclusion section has been rewritten. The changes have been carried out between lines 397- 404 (page 16). We are very thankful with the improvements proposed by the reviewer.

  1. B. Point 4. The references are not in the format indicated by the journal: You can check the proper format in the instructions for authors section (https://www.mdpi.com/journal/jcm/instructions)
  2. Response 4. The format of the references was indeed wrong. Thank you for noticing. We have added the Journal of Clinical Medicine format to our Mendeley and it has been corrected.

Reviewer 2 Report

I am reviewing the article “EFFECTS OF PERIPHERAL ELECTROMAGNETIC FIELDS ON SPASTICITY. A SYSTEMATIC REVIEW.". The manuscript under consideration is an interesting article on an important topic. However, there are a few minor concerns.

1. The argument in the introduction is weak. The need to evaluated an effects of electromagnetic fields is not justified.

2. Sensory-motor disorder are also present in patients with spastic cerebral palsy, and should be addition in the introduction.

3. Did this study investigate only a few diseases?

4. Please let me know if the final selected paper did not have Botox injections or intrathecal baclofen therapy in combination.

5.  How the present study is different from previous systenatic review ones? Explain the novelty of your work.

Author Response

ITEMIZED LIST OF THE REVIEWERS COMMENTS

Manuscript ID:   jcm-1746312

Title: “EFFECTS OF PERIPHERAL ELECTROMAGNETIC FIELDS ON SPASTICITY. A SYSTEMATIC REVIEW”

Dear Reviewer,

We greatly appreciate the editor´s and reviewers’ kind and encouraging comments about our study. We have followed their suggestions, trying to incorporate them into the revised version of our manuscript. We uploaded the tracked changes manuscript, the clean version revised manuscript and itemized point-by-point response to the reviewer’ comments are presented below.

Editor´s and Reviewers´ comments:

                                                     *Reviewer 2

Response to reviewer 2 Coments.

Comments and Suggestions for Authors
I am reviewing the article “EFFECTS OF PERIPHERAL ELECTROMAGNETIC FIELDS ON SPASTICITY. A SYSTEMATIC REVIEW.". The manuscript under consideration is an interesting article on an important topic. However, there are a few minor concerns.

  1. Point 1. The argument in the introduction is weak. The need to evaluated an effects of electromagnetic fields is not justified.
  2. Response 1. Thanks to the reviewer for this concern. Introduction section has been modified with the objective of make our manuscript more comprehensive. In this aspect, the justification of the need for an assessment of the effects of electromagnetic fields has been introduced between lines 46-49 (page 2).  We wanted to justify its study because it has important biological effects that may be beneficial in spasticity without adverse effects.

  1. Point 2. Sensory-motor disorder are also present in patients with spastic cerebral palsy, and should be addition in the introduction.
  2. Response 2. In page 2, line 44, it was indicated that sensory-motor disorders were observed in different neurological pathologies, but we have added cerebral palsy, given its importance, as the reviewer 2 so aptly comments.
  3. Point 3. Did this study investigate only a few diseases?
  4. Response 3. The PICOS (Population, Intervention, Comparison, Outcomes and Study design) model was used to establish the inclusion criteria: (I) Population: humans with spasticity; (II) Intervention: Treatment with electromagnetic fields administered to the lower or upper limbs or spine; (III) Comparison: placebo, no treatment, a different electrotherapy modality or any other intervention; (IV) Outcomes: related to spasticity; (V) Study design: controlled clinical trials.

Taking these criteria into account, our search was as open as possible to any type of disease with spasticity in which electromagnetic pulses were used as a treatment, and given the small number of articles, only the specified diseases were found in the scientific literature. It was found in spinal cord injury in rats and cerebral palsy, but they were not clinical trials, but case studies.

It would be desirable that quality clinical trials are also conducted for other diseases. In fact, in the discussion, a paragraph was written to this effect: Page 14, line 305-308:

“It would be advisable to study the effect of magnetic fields in other diseases where prevalence is also high, such as spinal cord injuries where it is estimated that between 40% and 78% have spasticity  or in cerebral palsy where the percentage is even higher (72-91%)”.

  1. Point 4. Please let me know if the final selected paper did not have Botox injections or intrathecal baclofen therapy in combination.
  2. Response 4.

We checked all the articles to find out if such therapies were mentioned  and none of the articles mentioned botox injections. As for the intrathecal baclofen therapy, the article “Treatment of spasticity with magnetic stimulation; a doublé-blind placebo-controlled study”  (Nielsen et al, 1997) mentioned: …Antispastic medication with baclofen, tizanidin, and diazepam as well as functional electric peroneal nerve stimulation was discontinued one week before the study.

These same authors in the discussion section mentioned: “Pharmacotherapy has a similar clinical success rate as repetitive magnetic stimulation. Sachais et al reported in a controlled multicenter trial in patients with multiple sclerosis a reduction in clinical muscle tone score by 15% at a daily dosage of 70-80 mg baclofen." From and Heltberg in a double blind trial observed a reduction in Ashworth score by 28% and 29% after 4 weeks of treatment with baclofen (30-120 mg/daily and diazepam (10-40 mg/daily), respectively. In a double-blind crossover study,  Feldman et al,  found a reduction in resistance to passive movement in 65% of multiple sclerosis patients after 4 weeks of treatment at a daily dosage of 80 mg baclofen. Many multiple sclerosis patients with light to moderate spasticity can not tolerate or refuse pharmacological treatment because of side-effects including drowsiness, dizziness, nausea, and muscle weakness. It is our experience that this new treatment technique is well-tolerated by the patients and have no serious side-effects according to the patients. To offer this form of treatment it should be safe”.

In another article “Effects of Para-Spinal Repetitive Magnetic Stimulation on Multiple Sclerosis Related Spasticity”  (Serag 2014), the authors commented in the introduction section on the pharmacological treatment that can be used in spasticity, including intrathecal baclofen. : “ Management of spasticity includes drug treatments , botulinum toxin injection for focal spasticity , and intrathecal baclofen . Non pharmacological management is also considered, and this includes physical therapy, transcutaneous electric nerve stimulation , and non-invasive brain stimulation. The article they referenced in this regard was the following:

Borrini L, Bensmail D, Thiebaut JB, Hugeron C, Rech C, Jourdan C. Occurrence of adverse events in long-term intrathecal baclofen infusion: a 1-year follow-up study of 158 adults. Arch Phys Med Rehabil. 2014 Jun;95(6):1032-8. doi: 10.1016/j.apmr.2013.12.019. Epub 2014 Jan 6. PMID: 24407102.

  1. Point 5. How the present study is different from previous systenatic review ones? Explain the novelty of your work.
  2. Response 5.

Systematic reviews of peripherally applied electromagnetic pulses have been conducted on musculoskeletal pathology but not on spasticity. We found a pilot study  “Repetitive Peripheral Magnetic Stimulation as Pain Management Solution in Musculoskeletal and Neurological Disorders – A Pilot Study” (Dragana Zarkovic,  Krasimira Kazalakova, 2016)  studying the effect of peripheral electromagnetic fields, in which musculoskeletal and neurological pathology were mixed, but it was not included in our review because it did not meet our eligibility criteria. On the other hand, with regard to spasticity and electromagnetic field studies, there are several systematic reviews such as:

           McIntyre, A.; Mirkowski, M.; Thompson, S.; Burhan, A.M.; Miller, T.; Teasell, R. A Systematic Review and Meta-Analysis on the Use of Repetitive Transcranial Magnetic Stimulation for Spasticity Poststroke. PM R 2018, 10, 293–302.

           Kumru, H.; Murillo, N.; Vidal Samso, J.; Valls-Sole, J.; Edwards, D.; Pelayo, R.; Valero-Cabre, A.; Tormos, J.M.; Pascual-Leone, A. Reduction of Spasticity with Repetitive Transcranial Magnetic Stimulation in Patients with Spinal Cord Injury. Neurorehabil. Neural Repair 2010, 24, 435–441, doi:10.1177/1545968309356095.

           Fisicaro, F.; Lanza, G.; Grasso, A.A.; Pennisi, G.; Bella, R.; Paulus, W.; Pennisi, M. Repetitive Transcranial Magnetic Stimulation in Stroke Rehabilitation: Review of the Current Evidence and Pitfalls. Ther. Adv. Neurol. Disord. 2019, 12.

 However the application was transcranial and not at the peripheral level. Therefore,

our review is more novel in this sense because it studies the effects of peripherally

applied electromagnetic fields on spasticity.

Please, do not hesitate to contact me, if you require further corrections and information.

Thank you in advance

Round 2

Reviewer 1 Report

The points noted in the manuscript have been properly addressed.